# Quantum Circuit Optimization for Solving Discrete Logarithm of Binary Elliptic Curves Obeying the Nearest-Neighbor Constrained

**DOI:** 10.3390/e24070955

**Published:** 2022-07-09

**Authors:** Jianmei Liu, Hong Wang, Zhi Ma, Qianheng Duan, Yangyang Fei, Xiangdong Meng

**Affiliations:** 1State Key Laboratory of Mathematical Engineering and Advanced Computing, Zhengzhou 450001, China; jianmeiliu2022@outlook.com (J.L.); qhduan@meac-skl.cn (Q.D.); feiyangyang@pku.edu.cn (Y.F.); xiangdongmeng@meac-skl.cn (X.M.); 2Henan Key Laboratory of Network Cryptography Technology, Zhengzhou 450001, China

**Keywords:** elliptic curve, discrete logarithm, quantum circuit

## Abstract

In this paper, we consider the optimization of the quantum circuit for discrete logarithm of binary elliptic curves under a constrained connectivity, focusing on the resource expenditure and the optimal design for quantum operations such as the addition, binary shift, multiplication, squaring, inversion, and division included in the point addition on binary elliptic curves. Based on the space-efficient quantum Karatsuba multiplication, the number of CNOTs in the circuits of inversion and division has been reduced with the help of the Steiner tree problem reduction. The optimized size of the CNOTs is related to the minimum degree of the connected graph.

## 1. Introduction

The security of Elliptic Curve Cryptosystems is based on the difficulty of solving the discrete logarithm problem in an elliptic curve group. It seems more difficult to deal with the problem for solving discrete logarithm in $F_{2^{n}}$ than in $F_{p}$. The key agreement represents the protocol in which two or more parties together generate a secret key using a public channel [1,2,3]. For instance, better security can be achieved in Diffie–Hellman Key exchange by choosing a suitable elliptic curve in $F_{2^{155}}$ than in $F_{p}$ when *p* has 512 bits. The efficiency of the optimization for elliptic curve cryptosystems relies on the speed of the operations in the elliptic curve, whose core operation is the point addition. The efficient algorithms for elliptic curve cryptography are classified into high-level algorithms and low-level algorithms, i.e., group operations of elliptic curves and arithmetic operations in the fundamental finite field. Obviously, both of the above two-level operations should be optimized in order to realize the elliptic curve cryptosystem effectively.

With the intrinsic advantages in executing certain matrix multiplication operations, quantum algorithms are proposed to enhance data analysis techniques under some circumstances [4]. The first paper to discuss in detail how to use a quantum algorithm to solve elliptic curve discrete logarithm problem is by Proos and Zalka [5]. Based on this study, in 2017, Rötteler, Naehrig, Svore, and Lauter presented a concrete quantum resource estimation and the explicit quantum circuit for operations of point additions for solving the discrete logarithm problem in elliptic curves over $F_{p}$ [6].

While there is some common ground between the prime-field case and the characteristic-two case, there are also important differences. Elliptic curves over finite fields $F_{2^{n}}$ play a prominent role in modern cryptography. Published quantum algorithms dealing with such curves build on a short Weierstrass form in combination with affine or projective coordinates. Amento, Rötteler, and Steinwandt use projective coordinates to avoid divisions [7]. They need only 13 multiplications every step, which would result in $26n^{\log(3)+1}$ as the leading term in their Toffoli gate count if the multiplications were implemented using the space-efficient quantum Karatsuba multiplication [8]. Amento et al. show in their paper [7] the choice of how to represent the elements of $F_{2^{n}}$ can have a significant impact on the resource requirements for quantum arithmetic. In particular, they show how the Gaussian normal basis representations and “ghost-bit basis” representations can be used to implement inverters with a quantum circuit of depth O(nlog(n)). This is the first construction to compute inverse in $F_{2^{n}}^{*}$ with subquadratic depth reported in the literature.

The quantum circuit of computing inverse in $F_{2^{n}}^{*}$ in [7] is based on the Itoh–Tsujii algorithm [9] which exploits the property that, in a normal basis representation, squaring corresponds to a permutation of the coefficients. Because the map $\xi \to \xi^{2^{i}}$ is a bijection in $F_{2^{n}}^{*}$, it corresponds to an *n* by *n* nonsingular matrix, and all the elements in the matrix belong to $F_{2}$. Then, using an LUP-decomposition of this matrix, the needed exponentiation can be realized with $n^{2}+n$ CNOT gates in depth 2n. For i≥0 they define $\beta =\alpha^{2^{i}-1}$. Then the goal is to find $\alpha^{-1}={\left(\beta_{n-1}\right)}^{2}$ from $\beta_{1}=\alpha$. For this they exploit that $\beta_{i+j}=\beta_{i}\cdot \beta_{j}^{2^{i}}$ for all i,j≥0. Thus, in a polynomial basis representation, one evaluation of $\beta_{i+j}=\beta_{i}\cdot \beta_{j}^{2^{i}}$ can be realized in depth O(n) using $n^{2}$ Toffolis and $2n^{2}+n-1$ CNOT gates.

However, this use of projective coordinates has two disadvantages. First, they use many ancillary qubits and separate input and output qubits, leading to 10n qubits in one point-addition step even with space-efficient quantum Karatsuba multiplications. Second, projective coordinates have a much larger space disadvantage not pointed out in Ref. [7]. Furthermore, Ref. [7] does not specify the entirety of Shor’s algorithm, leaving open how exactly the presented results would be combined.

Building on the Karatsuba multiplier, the multiplication algorithm presented by Ref. [8] can be realized using $O(n^{\log(3)})$ Toffoli gates and 3n qubits, which has been exploited by Ref. [10]. However, there exists a disadvantage in the method of [8]. There are so many CNOT gates needed in Ref. [8], which is $O(n^{2})$.

The number of qubits and the connectivity between qubits in practical quantum devices are limited by the noisy environment. However, the resource costs have not been discussed in Refs. [5,6,7,8,9,10] when the quantum bit connectivity is limited. We discuss the quantum circuit optimization for solving discrete logarithm of elliptic curve in $F_{2^{n}}$, obeying the nearest-neighbor constrained. It has been shown that when operating a CNOT gate between two qubits, the number and the depth of CNOT gates needed are determined by the distance between the two qubits. Therefore, the number and the depth of CNOT gates needed in elementary operations (such as additions, binary shifts, multiplications, and squarings) for point additions are dominated by the arrangement of qubits. In this paper we treat division by a field element as multiplication by the inverse of that element and the inversion step is based on Fermat’s little theorem (i.e., using the Itoh–Tsujii algorithm to compute the inverse). With the help of the Steiner tree problem reduction in Refs. [11,12], we optimize the number of CNOT gates included in the point addition on binary elliptic curves under a constrained connectivity. The optimized size of the CNOTs is $O(n^{2}/\log\;\delta )$, where δ is the minimum degree of the connected graph. Based on this, for both division algorithms, the FLT-based algorithm preserves the similar number of Toffoli gates and qubits and suppresses the disadvantage previously in Ref. [10], which has roughly twice the number of the CNOT gate count compared with the GCD-based algorithm.

## 2. Materials and Methods

Each addition in $F_{2}$ takes one CNOT gate. The addition of two polynomials f(x),g(x) of degree at most n−1 takes *n* CNOT gates with depth 1. Considering the connectivity of qubits [13], four CNOT gates will be needed in performing a CNOT gate between the first qubit and the third qubit, which is shown in the Figure 1. Eight CNOT gates will be needed in performing a CNOT gate between the first qubit and the fourth qubit, which is shown in Figure 2. Therefore, 4(n−2) CNOT gates will be needed in performing a CNOT gate between the first qubit and the *n*-th qubit.

Let the connectivity of qubits corresponding to the coefficients of f(x),g(x) be: $$f_{0}-g_{0}-g_{1}-f_{1}-f_{2}-g_{2}-g_{3}-\cdot \cdot \cdot -f_{n-3}-f_{n-2}-g_{n-2}-g_{n-1}-f_{n-1}.$$ Then the number of and the depth of CNOT gates needed in the addition of f(x) and g(x) are still *n* and 1, respectively. When these qubits are arranged in the following order $$f_{0}-f_{1}-f_{2}-\cdot \cdot \cdot -f_{n-2}-f_{n-1}-g_{0}-g_{1}-g_{2}-\cdot \cdot \cdot -g_{n-2}-g_{n-1},$$ the number of and the depth of CNOT gates needed in the addition of f(x) and g(x) are $4\left(n-1\right)\cdot n=4n^{2}-4n$ and $4\left(n-1\right)\cdot n-\left(n-1\right)=4n^{2}-5n+1$, respectively.

For polynomials in $F_{2}\left[x\right]$ multiplication by *x* is a shift of the coefficient vector. This requires no quantum computation by doing a series of swaps. In a finite field $F_{2^{n}}$ we want to multiply a polynomial g(x) of degree at most n−1 by *x* then by a modular reduction by a fixed irreducible weight-ω degree-*n* polynomial m(x). In general, we let ω be 3 or 5. As m(x) is irreducible, it always has coefficient 1 for $x^{0}$, so after a reduction by m(x) that qubit will be 1 and if no reduction takes place that qubit will be 0, which means the modular shift algorithm is always reversible. Considering the connectivity of qubits, when the Hamming weight of m(x) is ω=3 and $m\left(x\right)=x^{n}+x^{t}+1$ (1≤t<n), we let the connectivity of qubits corresponding to the coefficients of g(x) be:$$g_{0}-g_{1}-\cdot \cdot \cdot -g_{t-2}-g_{t-1}-g_{t+1}-g_{t+2}-\cdot \cdot \cdot -g_{n-2}-g_{n-1}-g_{t+2}.$$ Then the number of and the depth of CNOT gates needed in multiplying g(x) by *x* then by a modular reduction by m(x) are still *n* and 1, respectively. When these qubits are arranged in the following order
$$g_{0}-g_{1}-\cdot \cdot \cdot -g_{n-2}-g_{n-1},$$
the number of and the depth of CNOT gates needed in multiplying g(x) by *x* then by a modular reduction by m(x) are 4(n−t−1) and 4(n−t−1). respectively.

When the Hamming weight of m(x) is ω=5 and $m\left(x\right)=x^{n}+x^{t_{3}}+x^{t_{2}}+x^{t_{1}}+1$ ($1\leq t_{1}<t_{2}<t_{3}<n$), let the connectivity of qubits corresponding to the coefficients of g(x) be:$$g_{0}-g_{1}-\cdot \cdot \cdot -g_{n-2}-g_{t_{3}}-g_{n-1}-g_{t_{2}}-g_{t_{1}}$$
or
$$g_{0}-g_{1}-\cdot \cdot \cdot -g_{n-2}-g_{t_{3}}-g_{n-1}-g_{t_{1}}-g_{t_{2}}$$
or
$$g_{0}-g_{1}-\cdot \cdot \cdot -g_{n-2}-g_{t_{2}}-g_{n-1}-g_{t_{1}}-g_{t_{3}}$$ or $$g_{0}-g_{1}-\cdot \cdot \cdot -g_{n-2}-g_{t_{2}}-g_{n-1}-g_{t_{3}}-g_{t_{1}}$$ or $$g_{0}-g_{1}-\cdot \cdot \cdot -g_{n-2}-g_{t_{1}}-g_{n-1}-g_{t_{2}}-g_{t_{3}}$$ or $$g_{0}-g_{1}-\cdot \cdot \cdot -g_{n-2}-g_{t_{1}}-g_{n-1}-g_{t_{3}}-g_{t_{2}}.$$

Then the number of and the depth of CNOT gates needed in multiplying g(x) by *x* then by a modular reduction by m(x) are 4 and 3, respectively. When these qubits are arranged in the following order $g_{0}-g_{1}-\cdot \cdot \cdot -g_{n-2}-g_{n-1}$, the number of and the depth of CNOT gates needed in multiplying g(x) by *x* then by a modular reduction by m(x) are at most $2\left(n-t_{1}-1\right)+1+2\left(n-t_{1}-2\right)+1=4\left(n-t_{1}\right)-4$ and $2\left(n-t_{1}-1\right)+1+\left(n-t_{1}-2\right)+1=3\left(n-t_{1}\right)-2$, respectively. The number of and the depth of CNOT gates are at least $2\left(n-t_{3}-1\right)+1+2\left(n-t_{3}-2\right)+1=4\left(n-t_{3}\right)-4$ and $2\left(n-t_{3}-1\right)+1+\left(n-t_{3}-2\right)+1=3\left(n-t_{3}\right)-2$, respectively.

For multiplication, if we use a space-efficient Karatsuba algorithm by Van Hoof, we will need $O(n^{2})$ CNOT gates, $O(n^{\log(3)})$ Toffoli gates, and 3n total qubits: 2n qubits for the input f(x),g(x), and *n* separate qubits for the output f(x)·g(x). In a multiplication, most CNOT gates are needed in the processes of multiplying by $1+x^{k}$ or ${\left(1+x^{k}\right)}^{-1}$ where *k* has ⌈log(n)⌉ values and each process need $O(n^{2})$ CNOT gates. In the quantum algorithm for the division we have to use up to $2(k_{1}\;+\;t\;-\;1)$ multiplications, so 4(log$\left(n\right))\cdot O\left(n^{2}\right)\cdot \left(k_{1}\;+\;t\;-\;1\right)$ (i.e., $O(n^{2}(\log^{2}$(n)))) CNOT gates will be needed in the quantum algorithm for a division. If we take the constrained connectivity into consideration, at most 16(log$\left(n\right))\cdot O\left(n^{2}\right)\cdot \left(n-2\right)\cdot \left(k_{1}\;+\;t\;-\;1\right)$ (i.e., $O(n^{3}(\log^{2}$(n)))) CNOT gates will be needed.

If the irreducible polymomial is fixed to a trinomial $m\left(x\right)=x^{n}+x^{t}+1$ (1≤t<n) or a pentanomial $m\left(x\right)=x^{n}+x^{t_{3}}+x^{t_{2}}+x^{t_{1}}+1$ ($1\leq t_{1}<t_{2}<t_{3}<n$) each multiplying by $1+x^{k}$ or ${\left(1+x^{k}\right)}^{-1}$ will need about (log(n))·n CNOT gates. Then we use up to $2(k_{1}\;+\;t\;-\;1)$ multiplications in the quantum algorithm for the division. Therefore only about 4(log${\left(n\right))}^{2}\cdot n\cdot \left(k_{1}\;+\;t\;-\;1\right)$ CNOT gates are needed in the quantum algorithm for a division. When the constrained connectivity has been taken into consideration, at most 16(log${\left(n\right))}^{2}\cdot n\cdot \left(n-2\right)\cdot \left(k_{1}\;+\;t\;-\;1\right)$ CNOT gates will be needed.

Take for example the irreducible polynomial $m\left(x\right)=x^{4}+x+1$, based on which the finite field $F_{2^{4}}$ can be constructed. The quantum circuit of the space-efficient Karatsuba algorithm by Van Hoof is shown in the Figure 3:

The simulation is ran under IBM T-like graph (T65). The topological structure of IBM T65 is depicted below::$$f_{2}-g_{3}-g_{2}-g_{0}-h_{1}-f_{0}-f_{3}-f_{1}-h_{2}-h_{3}\;| | |\;\;\;\;\;\;\;\;\;\;\;\;\;\;\;h_{0}\;\;\;\;\;\;\;\;\;\;\;\;\;\;\;\;\;\;\;\;\;g_{1}$$

For the sake of optimizing the number and the depth of CNOT gates while preserving the similar number of Toffoli gates and qubits, we adopt the implementation of a Toffoli gate shown in Figure 4, which has been proposed by Ref. [14]. If we take the constrained connectivity into consideration, 812 CNOT gates will be needed in the quantum circuit for the space-efficient Karatsuba algorithm by Van Hoof.

Because the map $\xi \to \xi^{2^{i}}$ is a bijection in $F_{2^{n}}$, we can think of squaring in $F_{2^{n}}$ as a circuit that replaces the input with the result. To square and replace the input, we make use of the fact that squaring is a linear map and we can write that map as an *n* by *n* matrix. Using an LUP-decomposition, we get a lower triangular, upper triangular, and permutation matrix, which can be translated into a circuit consisting of at most $n^{2}-n$ CNOT gates and a number of swaps. In the quantum algorithm for the division we have to use up to 4n−4 squarings, so $4n^{3}-8n^{2}+4n$ CNOT gates will be needed in the quantum algorithm for a division. If we take the constrained connectivity into consideration, at most $16n^{4}-64n^{3}+80n^{2}-32n$ CNOT gates will be needed.

If the irreducible polymomial is fixed to a trinomial $m\left(x\right)=x^{n}+x^{t}+1$ (1≤t<n) or a pentanomial $m\left(x\right)=x^{n}+x^{t_{3}}+x^{t_{2}}+x^{t_{1}}+1$ ($1\leq t_{1}<t_{2}<t_{3}<n$), each squaring will need about 2n CNOT gates. Then we use up to 4n−4 squarings in the quantum algorithm for the division. Therefore, only about $8n^{2}-8n$ CNOT gates are needed in the quantum algorithm for a division. When the constrained connectivity has been taken into consideration, at most $32n^{3}-96n^{2}+64n$ CNOT gates will be needed.

Take for example the irreducible polynomial $m\left(x\right)=x^{4}+x+1$, based on which the finite field $F_{2^{4}}$ can be constructed. The quantum circuit of the squaring for a polynomial $a\left(x\right)=a_{0}+a_{1}x+a_{2}x^{2}+a_{2}x^{3}$ in $F_{2^{4}}$ need 5 CNOT gates. If we take the constrained connectivity into consideration, 8 CNOT gates will be needed.

## 3. Results

Fermat’s little theorem can be extended for binary finite fields to $f^{2^{n}-2}=f^{-1}\;\mathrm{mod}$ m(x) where *n* is the degree of m(x). With the help of squarings, this can be calculated in *n* multiplications and n−1 squarings: $f^{2^{n}-2}=f^{2}\cdot f^{2^{2}}\cdot f^{2^{3}}\cdot \ldots \cdot f^{2^{n-1}}$. Itoh and Tsujii give an improvement to this straightforward method to reduce the cost to below 2log(n) multiplications and n−1 squarings. The Itoh–Tsujii algorithm works as follows:(1)Write n−1 as $\left[k_{1},\ldots ,k_{t}\right]$ with $\sum_{s=1}^{t}2^{k_{s}}=n-1$ and $k_{1}>\ldots >k_{t}\geq 0$. Note that *t* is the Hamming weight of n−1 in binary and t≤⌊log(n−1)⌋+1 and $k_{1}=\lfloor$log(n−1)⌋;(2)Calculate $f^{2^{2^{k_{1}}}-1}$ with $k_{1}$ multiplications, and save the intermediate results $f^{2^{2^{k_{t}}}-1}$, $f^{2^{2^{k_{t-1}}}-1},\ldots ,f^{2^{2^{k_{1}}}-1}$;(3)Calculate $f^{2^{n-1}-1}=\left\{\ldots \left\{{\left(f^{2^{2^{k_{1}}}-1}\right)}^{2^{2^{k_{2}}}}\left(f^{2^{2^{k_{2}}}-1}\right)\right\}{}^{2^{2^{k_{3}}}}\ldots \right\}{}^{2^{2^{k_{t}}}}\left(f^{2^{2^{k_{t}}}-1}\right)$ using t−1 multiplications;(4)Square the result to get $f^{-1}$. In total, $k_{1}\;+\;t\;-\;1$ multiplications are needed for the inversion $f^{-1}\;\mathrm{mod}$
m(x). The quantum circuit of computing $f^{-1}\;\mathrm{mod}$$x^{4}+x+1$ is shown in Figure 5.

Therefore, $2n^{\log(3)}\left(k_{1}\;+\;t\;-\;\frac{1}{2}\right)$ Toffoli gates and n·max
$\left(k_{1}\;+\;t\;-\;1,k_{1}\;+\;1\right)$ ancillary qubits are needed for the division in the quantum case. The total number of logical qubits required for the division is 3n+n·max
$\left(k_{1}\;+\;t\;-\;1,k_{1}\;+\;1\right)$.

The classic algorithm for the inversion $f^{-1}\;\mathrm{mod}$
m(x) uses n−1 squarings and the quantum algorithm for the division has to use up to 4n−4.

Only CNOT gates exist in quantum circuits of squarings and, multiplying by $1+x^{k}$ or ${\left(1+x^{k}\right)}^{-1}$ in the multiplications, these circuits are CNOT circuits, which cost many CNOT gates.

For a graph G(V,E) with *n* vertices, without loss of generality, we assume that the degree of vertices are denoted as $d_{1}\leq d_{2}\leq \cdots \leq d_{n}$. A theorem has been given by Bujiao Wu et al. in [12], which optimizes the size of CNOTs.

Given a set of terminals and a connectivity graph, the algorithm performs breadth-first search outwards from each of the terminals. When the paths collide, the nodes along that path consolidate into a single node and all the edges adjacent to the consolidated nodes are placed adjacent to this new node. The process is restarted with this node as a new terminal. From many trials, it seems that this approximation is sufficient to see a large reduction in the CNOT count of the output circuit. The choice of Steiner tree approximation algorithm for this purpose depends on the user’s efficiency and performance requirements.

It follows that the optimized size in Theorem 1 is asymptotically tight for a nearly regular graph.

**Theorem** **1.** 
*Given connected graph G(V,E) with*

$$\sum_{i\leq k}d_{i}\geq n,$$

*then there is a polynominal time algorithm to construct an equivalent
$O(\frac{n^{2}}{log(n/k)})$ size CNOT circuit for any n-qubit CNOT circuit on topological graph G, and there needs at least
$\Omega (\frac{n^{2}}{log\;d_{n}})$ size of CNOT gates for some invertible matrix.*


We can see the proof of Theorem 1 in [12]. Let k=n/δ for any given CNOT circuits with *n* qubits under a constrained connectivity, in which δ is the minimum degree of the connected graph. Then it can be easily shown that the sum of degrees for any *k* vertices is greater than *n*. Therefore, we will get CNOT circuits who have $O(n^{2}/\log\;\delta )$ CNOT gates. Due to the lower bound of the size of CNOT gates being $\Omega (n^{2}/\log\;\delta )$ for any CNOT circuits on a connected graph [15], the bound $O(n^{2}/\log\;\delta )$ is tight for a regular graph. Let δ=4, then the size and the depth of CNOT gates needed in the quantum algorithm for the division will be cut in half.

## 4. Simulation of the Improved Quantum Circuit for Division Algorithm

In this paper, with the help of the Q# language, the resource estimation of the quantum circuit for the division algorithm used to solve discrete logarithms of elliptic curves in $F_{2^{n}}$ has been simulated. It has been shown that based on the space-efficient quantum Karatsuba multiplication, the number of CNOTs in the circuits of inversion and division has been reduced with the help of the Steiner tree problem reduction.

From Table 1, it can be seen that when the FLT-based algorithm is used for the division algorithm, the optimized quantum circuit of this paper is better in terms of the size and the depth of CNOT gates than that of [6]. Due to the space-efficient quantum Karatsuba multiplication, both the consumption of qubits and the consumption of Toffoli gates are also quite good.

Table 1 has also shown that the optimized quantum circuit of this paper where the FLT-based algorithm is used for the division algorithm is better in terms of the size of CNOT gates than that of [6], where the GCD-based algorithm is used for the division algorithm.

If the constrained connectivity has been taken into consideration, about $128n^{3}$ CNOT gates will be needed in the quantum circuit for the division algorithm proposed by this paper.

## 5. Discussion and Conclusions

With the development of time, extensive attention has been attracted by the field of quantum computation. The main tool for researching the implementation of quantum algorithms is quantum circuit models, whose optimization is a direction worthy of study. In this paper, we have comprehensively discussed the quantum circuit of solving discrete logarithms of elliptic curves in $F_{2^{n}}$ and have made further optimizations of the size and the depth of CNOT gates. Based on the space-efficient quantum Karatsuba multiplication, we have reduced the number of CNOTs in the circuits of inversion and division with the help of the Steiner tree problem reduction.

In the future, we will consider the quantum circuit optimizations of practical quantum devices in noisy environments and assess the performances of quantum algorithms on practical quantum devices.

## Figures and Tables

**Figure 1 entropy-24-00955-f001:**
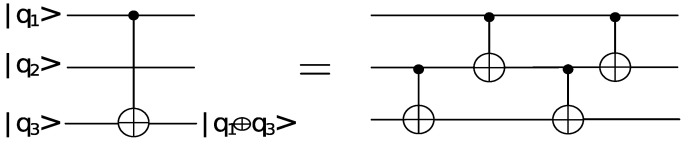
The quantum circuit of performing a CNOT gate between $q_{1}$ and $q_{3}$.

**Figure 2 entropy-24-00955-f002:**
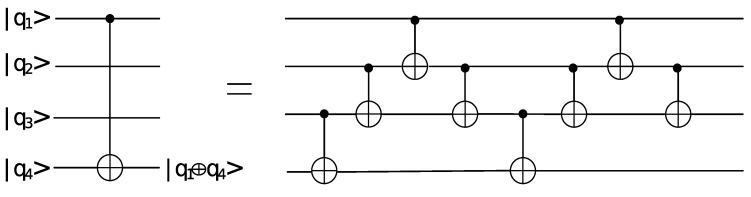
The quantum circuit of performing a CNOT gate between $q_{1}$ and $q_{4}$.

**Figure 3 entropy-24-00955-f003:**
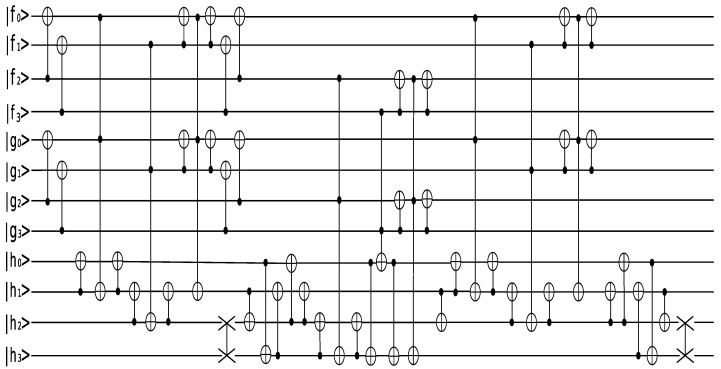
The quantum circuit of f(x)·g(x)mod $x^{4}+x+1$.

**Figure 4 entropy-24-00955-f004:**
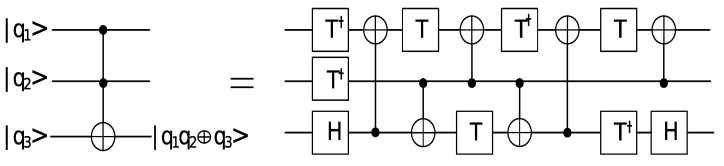
The quantum circuit of implementing a Toffoli gate.

**Figure 5 entropy-24-00955-f005:**
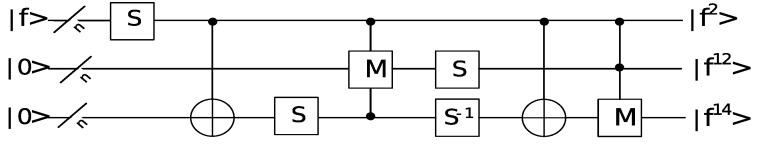
The quantum circuit of computing $f^{-1}\mathrm{mod}$$x^{4}+x+1$.

**Table 1 entropy-24-00955-t001:** Comparison of quantum resource of division algorithms.

*n*	QuantumCircuit	CNOT	Toffoli	Qubits	Depth
8	[6] for GCD-based	1516	3641	67	4113
8	[6] for FLT-based	2212	243	56	1314
8	This paper	1106	243	56	712
16	[6] for GCD-based	5072	10,403	124	12145
16	[6] for FLT-based	10,814	1053	144	5968
16	This paper	5407	1053	144	3265
127	[6] for GCD-based	227,902	277,195	903	378,843
127	[6] for FLT-based	502,870	50,255	1778	203,500
127	This paper	251,435	50,255	1778	105,989
163	[6] for GCD-based	375,738	442,161	1156	612,331
163	[6] for FLT-based	906,170	83,353	1956	451,408
163	This paper	453,085	83,353	1956	242,692
233	[6] for GCD-based	743,136	827,977	1646	1,172,733
233	[6] for FLT-based	1,486,464	132,783	3029	640,266
233	This paper	743,232	132,783	3029	344,230
283	[6] for GCD-based	1,088,400	1,202,987	1997	1,708,863
283	[6] for FLT-based	2,708,404	236,279	3962	1,434,686
283	This paper	1,354,202	236,279	3962	757,585
571	[6] for GCD-based	4,266,438	4,461,673	4014	6,494,306
571	[6] for FLT-based	10,941,536	814617	9136	6,151,999
571	This paper	5,470,768	814,617	9136	3,416,615

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
