# Peer review of "Quantum Circuit Optimization for Solving Discrete Logarithm of Binary Elliptic Curves Obeying the Nearest-Neighbor Constrained"

_entropy, 2022, doi:10.3390/e24070955_

Round 1
Reviewer 1 Report
See my report.

Reviewer 2 Report
I have carefully read the manuscript entropy-1771638. The Authors develop an optimization of the quantum circuit for the discrete logarithm of an elliptic curve in F2n under constrained connectivity, focusing on the optimal design for quantum operations such as the addition, binary shift, multiplication, squaring, inversion and division included in the point addition on binary elliptic curves. I think that the main contribution of the present manuscript is the reduction of CNOTs in the circuits of inversion and division with the help of the Steiner tree problem reduction, by using a space-efficient quantum Karatsuba multiplication. The Authors have probed that the optimized size of the CNOTs is O( n2/log δ ), where δ is the minimum degree of the connected graph.
There are some misprints, ie at line 61: However, the resource costs in case quantum bit connectivity is limited have not been discussed in [2–7].
I find the manuscript represents an interesting contribution to the field of quantum information. The motivation, the formalism and the results are presented in a clear manner.
From my previous observations, I consider that the present manuscript should be published as a regular article in Entropy after the spelling has been checked.
